# Adding Genetics to the Risk Factors Model Improved Accuracy for Detecting Visual Field Progression in Newly Diagnosed Exfoliation Glaucoma Patients

**DOI:** 10.3390/biomedicines12061225

**Published:** 2024-05-31

**Authors:** Marcelo Ayala

**Affiliations:** 1Department of Clinical Neuroscience, Institute of Neuroscience and Physiology, Sahlgrenska Academy, University of Gothenburg, 40530 Gothenburg, Sweden; marcelo.ayala@vgregion.se; 2Eye Department, Region Västra Götaland, Skaraborg Hospital/Skövde, 54142 Skövde, Sweden; 3Department of Clinical Neuroscience, Karolinska Institute, 17165 Stockholm, Sweden

**Keywords:** genetics, exfoliation glaucoma, visual fields, cohort studies, intraocular pressure, models

## Abstract

Background: This study aims to determine whether including genetics as a risk factor for progression will improve the accuracy of the models used in newly diagnosed exfoliation glaucoma patients. Methods: This was a prospective cohort study. This study included only patients who were newly diagnosed with exfoliation glaucoma and received treatment upon inclusion. Blood samples were taken from all patients at inclusion to test for the single nucleotide polymorphisms (SNPs) *LOXL-1* rs2165241 and rs1048661. Results: This study found that the frequency of SNPs, as well as intraocular pressure (IOP), mean deviation (MD), and visual field index (VFI) values at diagnosis, were significant predictors of visual field deterioration (*p* ≤ 0.001). This study showed that interaction terms, including SNPs, were highly significant (*p* ≤ 0.001). Furthermore, logistic regression analysis also showed highly significant results for interaction terms when SNPs were included (*p* ≤ 0.001). Finally, the area under the curve (AUC) analysis showed an increased value of around 10–20% when SNPs were included. Conclusions: Adding genetic factors to the well-known clinical risk factors can increase the accuracy of models for predicting visual field deterioration in exfoliation glaucoma patients. However, further studies are needed to investigate the role of other genes in this process.

## 1. Introduction

Glaucoma is a set of eye disorders that gradually damage the optic nerve and often cause permanent loss of the visual field. The condition usually results from high intraocular pressure (IOP) but can also occur with normal or low IOP. Glaucoma affects millions of people globally and is a major cause of blindness, which poses a significant public health challenge [1]. There are several types of glaucoma, with primary open-angle glaucoma being the most common worldwide. However, in Scandinavia, exfoliation glaucoma is the most widespread type [2]. Exfoliation glaucoma (EXFG) is a specific form of glaucoma that is characterized by the presence of exfoliation material in the anterior chamber of the eye. This material is often described as flaky, white deposits accumulating on different eye structures. This accumulation leads to increased IOP and damage to the optic nerve [3,4].

Exfoliation glaucoma is associated with the shedding of abnormal extracellular material in the eye’s anterior segment. Although the exact cause of exfoliation glaucoma is not fully understood, genetic factors have been demonstrated to contribute to its development. The Lysyl Oxidase Like 1 (*LOXL-1*) gene is the most studied gene related to EXFG [5]. The *LOXL-1* gene encodes for the lysozyme oxidase enzyme, involved in elastin biogenesis and collagen cross-linking [6]. In a previous article, the authors demonstrated the association between three single nucleotides (SNPs) and the presence of exfoliation glaucoma in a Swedish population [7]. Furthermore, two SNPs (rs2165241 and rs1048661) have been linked to an increased risk of visual field deterioration [8].

Several factors can contribute to the development of glaucoma. While some risk factors are beyond an individual’s control, it is essential to be aware of them to diagnose the condition early and manage it effectively. These risk factors include age, family history, ethnicity, increased IOP, and exfoliation [9,10,11]. However, simply having one or more of these risk factors does not necessarily mean that an individual will develop glaucoma. Conversely, some people may develop the condition despite having no apparent risk factors. As a result, individuals with one or more risk factors must undergo regular eye examinations to detect and manage glaucoma early. With early intervention, it may be possible to slow down or even prevent further vision loss associated with this condition.

This study aimed to evaluate the potential impact of incorporating genetics as a risk factor in the models utilized for newly diagnosed exfoliation glaucoma patients. This study is expected to produce valuable insights into the potential benefits of incorporating genetic factors into the exfoliation glaucoma diagnostic framework.

## 2. Materials and Methods

This study observed patients diagnosed with EXFG and was conducted as a non-randomized prospective cohort study. Patients who visited the Ophthalmology Department at Skaraborg Hospital, Skövde, Sweden, between 1 January 2012 and 31 December 2017 were included in this study. This study followed the Strengthening the Reporting of Observational studies in Epidemiology (STROBE) guidelines for reporting observational studies [12] (Appendix A). Before enrolling in this study, all participants were provided with both written and oral information. This study was approved by the Regional Ethical Committee at the University of Gothenburg (DN:119-12) and followed the principles of the Helsinki Declaration.

This study’s methodology was previously published and involved patients who were recently diagnosed with exfoliation glaucoma (EXFG) [13,14]. To summarize, all patients underwent an ophthalmological exam conducted by the same ophthalmologist (MA). At the time of diagnosis (inclusion), various variables were measured, including age, sex, IOP (using a Goldmann applanation tonometer), central corneal thickness (CCT), visual acuity (determined with the Snellen chart), gonioscopy, cup-to-disc ratio (C/D-ratio) of the optic nerve, and visual field using the Humphrey Field Analyzer (Carl-Zeiss, Straße 22, 73447 Oberkochen, Germany) using the 24-2 strategy of the Swedish Interactive Threshold Algorithm (SITA fast). Additionally, all patients completed a questionnaire about their medical history, which included diabetes, hypertension, smoking, medication intake, and family history of glaucoma.

As part of this study, all patients were given eye drops to decrease their IOP and maintain it at ≤20 mmHg. The patient’s visual acuity, IOP, medication count (i.e., eye drops), and visual field were evaluated at every appointment (every six months). The patients were closely monitored in accordance with the Swedish Guidelines for Glaucoma Care [15]. In cases where the desired IOP level could not be attained, the number of medications was incremented and/or selective laser trabeculoplasty (SLT) was performed.

### 2.1. Endpoints

This study’s primary objective was to ascertain the progression of the visual field employing highly reliable visual fields. The reliable visual fields were defined as those that displayed ≤15% false positives and fewer than 20% fixation losses. To ensure accuracy, only visual fields with an MD (mean deviation) between 0 and −16 dB were included. Including visual fields with significant damage can lead to “ceiling effects” and may not represent actual damage [16].

This study measured visual field progression using three parameters: mean deviation (MD), visual field index (VFI), and the guided progression analysis (GPA). MD estimates the overall level of visual field sensitivity in decibels (dB), with normal visual fields around 0 dB and significantly damaged visual fields around −20 dB. The VFI, on the other hand, is based on MD values but is more specific in detecting glaucoma damage. It emphasizes central parts of the visual field and is less affected by cataract development. The VFI calculates visual fields in percentages, with a normal visual field showing 100%.

The Guided Progression Analysis (GPA) system classifies the visual field progression as absent, possible, or likely. This study combined the possible and likely categories to form a single outcome of progression versus no progression. This simplified the outcome variable into a dichotomy of visual field progression.

### 2.2. Genetic Analysis

Each patient underwent venipuncture to extract blood samples as part of the inclusion process. Standard procedures were then followed to extract DNA from the samples. Subsequently, LGC Genomics (Hoddesdon, Herts, UK) utilized the KASPar PCR SNP genotyping system to genotype two specific SNPs (rs2165241 and rs1048661) in the *LOXL1* gene. The genotyping process yielded a success rate of over 95% for all SNPs and demonstrated adherence to the Hardy–Weinberg equilibrium. The genetic variants identified in the patients were analyzed using an additive genetic model, which was determined to be the most appropriate model for this study [7]. In this model, the non-risk homozygous genotype was assigned a value of 0, the heterozygous genotype was assigned a value of 1, and the high-risk homozygous genotype was assigned a value of 2.

### 2.3. Statistics

This study’s statistical analyses were performed using IBM’s SPSS software (29.0.1.1) (Armonk, NY 10504, USA). Initially, a single linear regression analysis was conducted on all predictor variables, and only statistically significant predictors (*p* ≤ 0.05) were selected for further analysis. The Kolmogorov–Smirnov test was used to assess the normality of these predictors, and Pearson’s coefficients were used to study correlations between them.

All significant predictors from the initial regression were utilized in the next step, where three strategies were applied with the same predictors. The first method involved a linear regression analysis to test the continuous outcome variables MD and VFI. The regression analysis also calculated the R^2^ value and “B-coefficient”. The model was adjusted for covariates. General Linear Models in SPSS were used to calculate interaction terms.

The second method was a logistic regression analysis for the binary outcome variable GPA, which also provided the R^2^ and the “Exp B coefficient”. Another accuracy value was shown by the regression analysis, namely the prediction accuracy (PA) from the “classification table”. All regression models and R^2^ values were adjusted for covariates.

The third method was a receiver operator curve with the area under the curve (AUC). The AUC was calculated based on the results from the logistic regression analysis. A predictive curve for interactions between predictors was calculated based on probability calculations from the logistic regression analysis.

Lastly, a power analysis for MD values based on linear regression was conducted. It revealed that a sample size of at least 81 experimental subjects was necessary to detect a 10% difference in MD over three years with a power of 0.80 and a significance level of 0.05.

## 3. Results

In this study, a total of 96 patients were included. Regrettably, 16 patients had to be excluded from this study as they did not meet the inclusion criteria. The reasons for exclusion varied, ranging from having undergone glaucoma surgery to poor adherence to check-up visits and low-quality visual fields. These exclusions were necessary to ensure this study’s accuracy and reliability, as factors such as glaucoma surgery could have influenced this study’s outcome. At the same time, poor adherence to check-up visits and low-quality visual fields could have affected the reliability of the results.

The cohort’s general characteristics were previously published [13,14]. To summarize, the patients included in this study were approximately 70 years old, with an almost equal distribution of male and female participants. Their visual acuity upon inclusion averaged 0.8 (Snellen units), and their mean IOP was relatively high at 32.52 mmHg. Out of the 96 patients, 66 had EXFG in one eye only, while 30 had it in both eyes. By the six-month check-up, the IOP values had decreased to 21.19 mmHg, and at the last IOP control three years later, the IOP was at 17.94 mmHg.

Upon inclusion, a significant correlation was found between IOP and MD values, as indicated by the Pearson’s coefficient (*p* = 0.001, r = 0.51). Similar results were observed for the correlation between IOP and VFI values at inclusion (Pearson’s coefficient: *p* = 0.001, r = 0.55). The correlation between MD and VFI values at inclusion was high (Pearson’s coefficient; *p*-, r = 0.92). Moreover, Pearson’s test was used to examine the correlation between two SNPs, *LOXL1*_rs2165241 and *LOXL1*_rs1048661, which revealed a significant correlation between the two SNPs (*p*-, r = 0.66).

We conducted an analysis of all variables using univariate linear regression for the endpoints, MD and VFI, at the end of this study. Our results revealed that several variables exhibited significant results, including SNPs, IOP, MD, and VFI at diagnosis, the number of medications, and SLT treatment during the three-year follow-up. Additionally, predictors IOP, MD, and VFI at diagnosis were normally distributed according to the Kolmogorov–Smirnov test (with *p*-values of 0.10, 0.17, and 0.25, respectively). To further validate our findings, we retested the predictors in a multivariate analysis with MD and VFI at the end of this study as endpoints. Even in the multivariate analysis, the predictors (SNPs, IOP, MD, and VFI at diagnosis) remained significant predictors for visual field deterioration. However, we observed that the interaction terms of IOP*MD at diagnosis and IOP*VFI at diagnosis were not significant. Conversely, the interaction terms, including IOP, MD, and VFI at diagnosis and the SNPs, were all highly significant. For additional information, please refer to Table 1.

A logistic regression analysis was performed to investigate the relationship between GPA results during the three-year follow-up period and various variables, including IOP, MD, and VFI at diagnosis, as well as the two specific SNPs (*LOXL1*_rs2165241 and *LOXL1*_rs1048661). The results indicated that all these factors remained significant predictors of GPA outcomes. Interestingly, the two SNPs exhibited different levels of accuracy, with *LOXL1*_rs2165241 showing better performance than *LOXL1*_rs1048661 (R^2^ values of 0.43 and 0.19, respectively) and higher prediction accuracy (PA values of 76% and 65%, respectively).

Although the interaction terms of IOP*MD at diagnosis were nearly significant (*p* = 0.04), the interaction term of IOP*VFI at diagnosis was not significant (*p* = 0.24), likely due to the presence of collinearity between predictors. Nevertheless, all interaction terms, including those related to the SNPs, showed high levels of significance and improved accuracy. Please see Table 2 for further details and information.

During the analysis, we used the area under curve (AUC) to evaluate the performance of our predictors. All the AUCs showed significant results. At the time of diagnosis, the predictors IOP, MD, and VFI had AUC values of 0.70, 0.80, and 0.79, respectively, indicating a good predictive performance. The AUC values increased by around 10–15% when genetics were included. For more information, please refer to Table 3.

The evaluation of AUCs showed significant outcomes for all predictors, with the predictive curve (interaction term) surpassing each individual predictor. We utilized IBM’s SPSS software (29.0.1.1) (Armonk, NY 10504, USA) to plot the data on a graph, generating Figure 1, Figure 2, Figure 3, Figure 4, Figure 5 and Figure 6 that depict these curves.

## 4. Discussion

In this study, we evaluated visual field deterioration in patients with newly diagnosed exfoliation glaucoma using three different strategies: MD, VFI, and GPA. The models showed increased accuracy when genetics were included as a risk factor. This finding was consistent across all three strategies and was observed in both linear and logistic regression models. Additionally, the AUC analysis confirmed these results.

This study found that clinical predictors such as IOP/MD and VFI at diagnosis were highly associated with glaucoma progression. This association was present in both linear and logistic regression analyses. However, when interaction terms were built using IOP and MD or VFI at diagnosis as predictors, the association was not significant. In the case of linear regression, the *p*-values were 0.89 and 0.55. In the case of logistic regression, a slight significance was found when MD was included in the interaction term (*p* = 0.04), but this was not observed when VFI was included (*p* = 0.24). These non-significant results may be due to collinearity between the predictors. The Pearson’s coefficient confirmed this, showing a significant correlation between IOP and MD (*p* = 0.001, r = 0.51). Similar results were observed for the correlation between IOP and VFI values at the time of inclusion (Pearson’s coefficient: *p* = 0.001, r = 0.55). In summary, this study suggested that patients diagnosed with high IOP tended to have more advanced visual field damage than those with low IOP.

When evaluating the accuracy of linear regression models, it is essential to consider the R^2^ value. This value measures the percentage of variance in the effect variable that can be explained by the predictor variable. A higher R^2^ value indicates a better fit. However, there is no defined cut-off value, so it is crucial to interpret R^2^ values clinically. In our study, we calculated adjusted R^2^ values due to multiple predictor variables in the multivariate linear regression. The IOP at diagnosis resulted in R^2^ values of around 0.34–0.37, while MD and VFI were around 0.55–0.58. The SNPs showed R^2^ values of 0.41–0.44. However, when we combined clinical predictors with SNPs in interaction terms, we saw an increase in the adjusted R^2^ values to 0.78–0.89, almost doubling the model accuracy. We also saw an increase in the R^2^ values for logistic regression analysis when combining SNPs with clinical predictors.

According to the logistic regression analysis, including SNPs increased predictive accuracy (PA). *LOXL1*_rs2165241, combined with MD/VFI at diagnosis, showed the highest PA values of 80% and 78%. On the other hand, when the other SNP (*LOXL1*_rs1048661) was included, PA values of 74% and 72% were obtained. PA values increased around 5–10% when the SNPs were added to the model. The AUC analysis also indicated a good prediction accuracy for all the predictors. The combination terms of MD/VFI * *LOXL1*_rs2165241 showed the highest AUC values (around 0.90), demonstrating an excellent prediction capacity. However, when *LOXL1*_rs1048661 was included, the AUC showed lower values (around 0.80), indicating that *LOXL1*_rs2165241 was a better predictor than *LOXL1*_rs1048661. It is worth noting that the two SNPs were highly correlated with each other (Pearson’s test, *p*-, r = 0.66), suggesting that they are part of the same genetic signal. Previous studies have reported high linkage disequilibrium (LD) between *LOXL1*_rs2165241 and *LOXL1*_rs1048661 [17]. The present study revealed a significant correlation between *LOXL1*_rs2165241 and *LOXL1*_rs1048661, indicating that they were part of the same genetic signal. Based on the results, it is likely that both SNPs need not be included in clinical models. The research further indicates that *LOXL1*_rs2165241 has the highest AUC values among the SNPs and, therefore, would be the most appropriate option for inclusion in the models.

This study examined two distinct SNPs, namely *LOXL1*_rs2165241 and *LOXL1*_rs1048661, which were selected based on previous research that linked them to visual field progression in patients with exfoliation glaucoma [8]. Another SNP, *LOXL1*_rs3825942, was identified in earlier studies involving Scandinavian populations which was strongly linked to exfoliation and exfoliation glaucoma [17,18]. Prior research investigated allele frequencies between healthy individuals and exfoliation glaucoma patients (case-control studies), revealing highly skewed frequencies of *LOXL1*_rs3825942. Specifically, the A allele of *LOXL1*_rs3825942 was found in only 0.4% of glaucoma patients but 17.9% of healthy individuals, consistent with previous findings [7,17]. Due to the limited sample size and low frequency of *LOXL1*_rs3825942 in glaucoma patients, it was not included in the current study.

Our study aimed to evaluate the progression of glaucoma in patients using visual fields, which is widely regarded as the most accurate method. We selected two parameters to assess the outcomes—mean deviation (MD) and visual field index (VFI). Although MD is a traditional technique that allows for comparison with previous studies, it has limitations as other eye conditions apart from glaucoma may also impact the MD values. For example, cataract development alters the MD values. Previous studies have evaluated visual field deterioration using the “pointwise” linear regression method [19,20]. The pointwise method correlates well with the VFI method [21]. The reason for evaluating visual field deterioration using VFI in our study is clinical. In Sweden, most ophthalmologists prefer using VFI to evaluate glaucoma progression as it is a more effective measure. VFI is a recalculation of the MD that emphasizes the central portions of the visual fields that are primarily affected by glaucoma [22].

It should be noted that this study has certain limitations to be considered. Firstly, this study only included patients diagnosed with MD between 0 and −16 dB, excluding cases of advanced glaucoma to avoid “ceiling effects” in the visual fields. Therefore, this study’s findings are only applicable to early and moderate glaucoma patients according to Hodapp’s classification [23]. Secondly, this study only involved patients with exfoliation glaucoma, limiting the generalizability of the results to other types of glaucoma. Thirdly, all patients in this study were recently diagnosed with glaucoma, meaning that disease progression was high, although the treatment received would slow down the disease. Fourthly, all patients included in this study were born in Scandinavia, and, given the significant role of genetics in exfoliation glaucoma, other ethnic groups may respond differently. Fifthly, this study only examined the *LOXL_1* gene and did not consider other genes or other SNPs located in the *LOXL_1*. Sixthly, the analysis conducted in this study cost around USD 200 per patient, totaling USD 20000. This study took place in Sweden, a wealthy country. Although the prices for genetic analysis are decreasing, they may still be unaffordable in poorer countries. Finally, disease progression was evaluated solely using visual fields, without anatomical measurements like optical coherence tomography (OCT).

## 5. Conclusions

The present study investigated the effectiveness of three distinct strategies for evaluating visual field deterioration. To achieve this, this study focused on analyzing the impact of genetics on the accuracy of risk factor models. Through the analysis of two specific genetic factors, *LOXL1*_rs2165241 and *LOXL1*_rs1048661, this study found that including genetic information significantly increased the accuracy of risk factor models. Furthermore, this study recommends that future research explore other genes related to EXFG to provide a more comprehensive understanding of the factors contributing to visual field deterioration. By expanding our knowledge of the genetic factors, we can better diagnose and manage visual field deterioration, improving the quality of life for individuals affected by this condition.

## Figures and Tables

**Figure 1 biomedicines-12-01225-f001:**
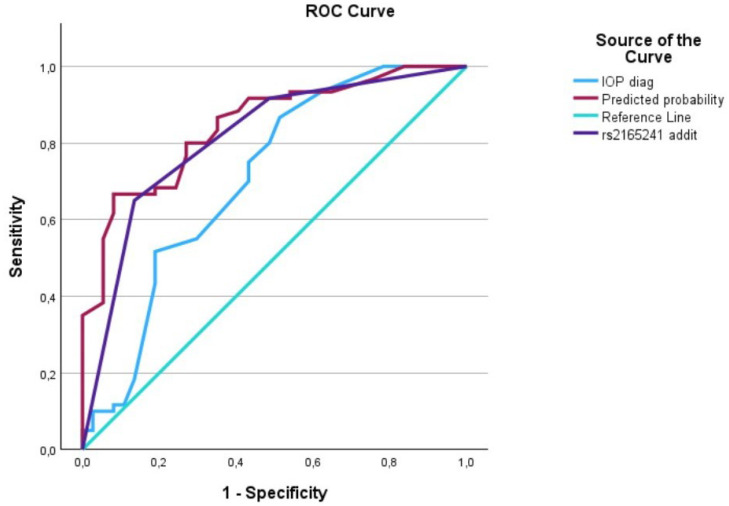
The AUC analysis of IOP at diagnosis and *LOXL1*_rs2165241. The IOP at diagnosis (---) was 0.70 [0.59–0.82], the *LOXL1*_rs2165241 (---) was 0.80 [0.72–0.90], and the interaction between IOP at diagnosis and *LOXL1*_rs2165241 (predicted curve) (---) was 0.85 [0.77–0.92].

**Figure 2 biomedicines-12-01225-f002:**
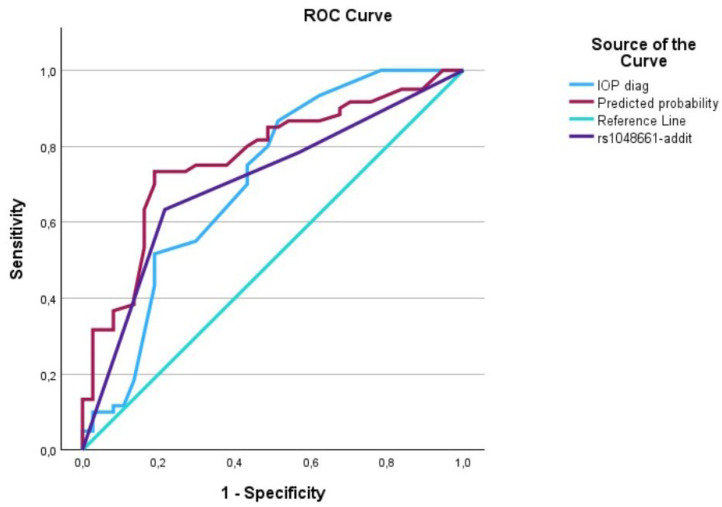
The AUC analysis of IOP at diagnosis and *LOXL1*_rs1048661. The IOP at diagnosis (---) was 0.70 [0.59–0.82], the *LOXL1*_rs1048661 (---) was 0.71 [0.60–0.81], and the interaction between IOP at diagnosis and *LOXL1*_rs1046661 (predicted curve) (---) was 0.77 [0.67–0.86].

**Figure 3 biomedicines-12-01225-f003:**
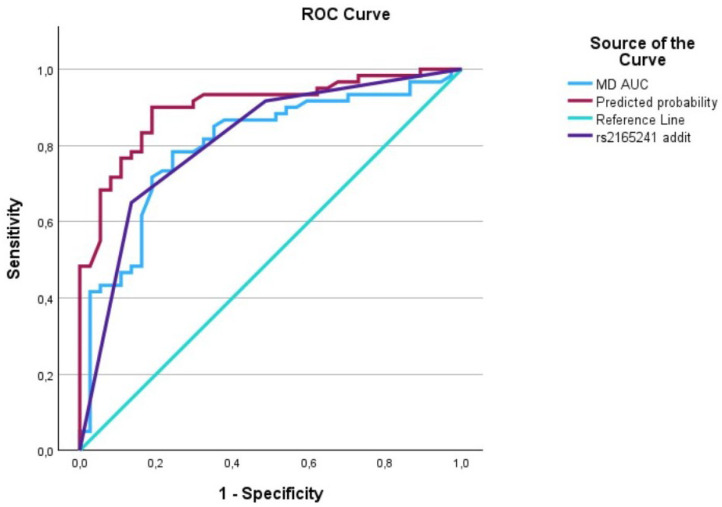
The AUC analysis of MD at diagnosis and *LOXL1*_rs2165241. The MD at diagnosis (---) was 0.8 [0.7–0.89], the *LOXL1*_rs2165241 (---) was 0.80 [0.72–0.90], and the interaction between IOP at diagnosis and *LOXL1*_rs2165241 (predicted curve) (---) was 0.90 [0.84–0.96].

**Figure 4 biomedicines-12-01225-f004:**
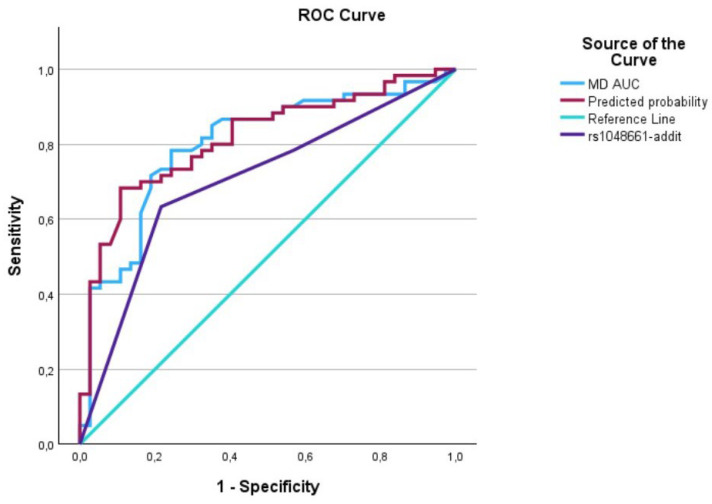
The AUC analysis of MD at diagnosis and *LOXL1*_rs1048661. The MD at diagnosis (---) was 0.8 [0.7–0.89], the *LOXL1*_rs1048661 (---) was 0.71 [0.60–0.81], and the interaction between MD at diagnosis and *LOXL1*_rs1046661 (predicted curve) (---) was 0.82 [0.73–0.90].

**Figure 5 biomedicines-12-01225-f005:**
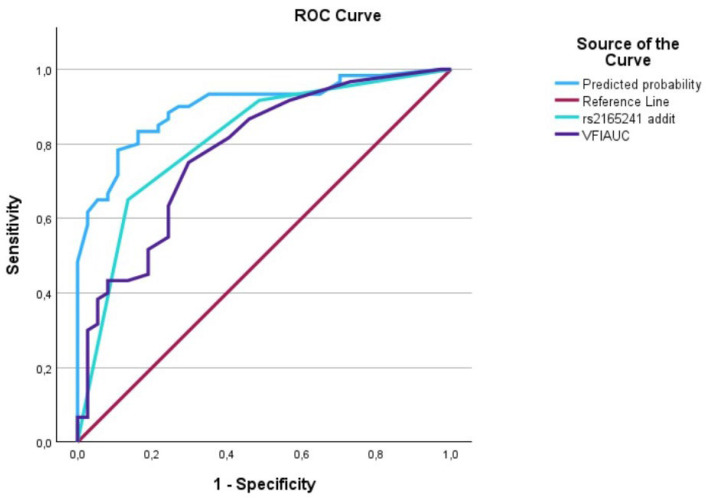
The AUC analysis of VFI at diagnosis and *LOXL1*_rs216524. The VFI at diagnosis (---) was 0.79 [0.69–0.88], the *LOXL1*_rs2165241 (---) was 0.80 [0.72–0.90], and the interaction between VFI at diagnosis and *LOXL1*_rs2165241 (predicted curve) (---) was 0.91 [0.85–0.97].

**Figure 6 biomedicines-12-01225-f006:**
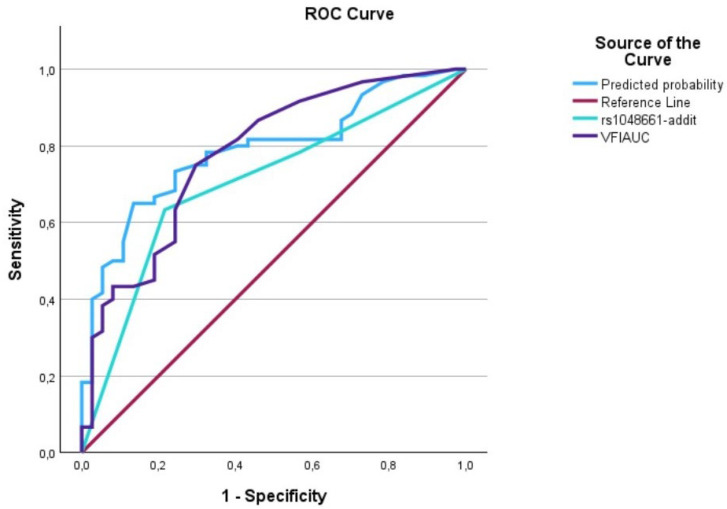
The AUC analysis of VFI at diagnosis and *LOXL1*_rs1048661. The VFI at diagnosis (---) was 0.79 [0.69–0.88], the *LOXL1*_rs1048661 (---) was 0.71 [0.60–0.81], and the interaction between VFI at diagnosis and *LOXL1*_rs2165241 (predicted curve) (---) was 0.81 [0.70–0.88].

**Table 1 biomedicines-12-01225-t001:** Multivariate analysis using the MD, VFI, and IOP values at diagnosis and the SNPs (*LOXL1*_rs2165241 and *LOXL1*_rs1048661) as predictors and MD and VFI at three years’ follow-up as outcomes.

	Outcomes	MD			VFI		
Predictors							
		B Coeff. [95% CI]	R^2 (1)^	*p*	B Coeff. [95% CI]	R^2 (1)^	*p*
**IOP at diagnosis ^1^**		−0.49 [−0.67, −0.31]	0.34	2 × 10^−3^ *	−1.19 [−1.74, −0.64]	0.37	4 × 10^−4^ *
**MD at diagnosis ^1^**		1.08 [0.98–1.19]	0.55	3 × 10^−3^ *	N.A.	N.A.	N.A.
**VFI at diagnosis ^1^**		N.A.	N.A.	N.A.	1.09 [1.01–1.17]	0.58	2 × 10^−4^ *
**LOXL1** **_rs2165241 ^1^**		0.78 [0.01–1.55]	0.41	0.01 *	1.04 [0.44–1.64]	0.43	6 × 10^−4^ *
**LOXL1** **_rs1048661 ^1^**		1.05 [0.25–1.85]	0.42	8 × 10^−3^ *	1.13 [0.52–1.75]	0.44	2 × 10^−4^ *
**IOP * MD at diagnosis ^2^**		N.A.	N.A.	0.89	N.A.	N.A.	N.A.
**IOP * VFI at diagnosis ^2^**		N.A.	N.A.	N.A.	N.A.	N.A.	0.55
**LOXL1** **_rs2165241 * IOP at diagnosis ^2^**		0.26 [0.12–0.34]	0.78	3 × 10^−6^ *	0.12 [0.08–0.14]	0.82	4 × 10^−7^ *
**LOXL1** **_rs1048661 * IOP at diagnosis ^2^**		0.31 [0.25–0.36]	0.79	2 × 10^−7^ *	0.22 [0.12–0.28]	0.83	5 × 10^−8^ *
**LOXL1_** **rs2165241 * MD at diagnosis ^2^**		0.82 [0.75–0.89]	0.85	6 × 10^−5^ *	N.A.	N.A.	N.A.
**LOXL1_** **rs1048661 * MD at diagnosis ^2^**		0.94 [0.88–0.99]	0.86	4 × 10^−6^ *	N.A.	N.A.	N.A.
**LOXL1_** **rs2165241 * VFI at diagnosis ^2^**		N.A.	N.A.	N.A.	1.06 [0.99–1.13]	0.88	3 × 10^−8^ *
**LOXL1_** **rs1048661 * VFI at diagnosis ^2^**		N.A.	N.A.	N.A.	1.08 [1.01–1.15]	0.89	4 × 10^−9^ *

MD: Mean deviation. VFI: Visual field index. IOP: Intraocular pressure. SLT: Selective laser trabeculoplasty. ^1^ Adjusted for SLT treatment and the number of medications. ^2^ Interaction terms were calculated using the General Linear Models strategy in SPSS. Also adjusted for SLT treatment and the number of medications. The ^(1)^ are the adjusted values so it’s the same than 1. The * are the significants values.

**Table 2 biomedicines-12-01225-t002:** Logistic regression analysis using the MD, VFI, and IOP values at diagnosis and the SNPs (*LOXL1*_rs2165241 and *LOXL1*_rs1048661) as predictors and GPA at three years’ follow-up as the outcome.

	Outcomes		GPA		
Predictors					
		Exp(B) Coeff. [95% CI]	R^2 (2)^	PA (%)	*p*
**IOP at diagnosis ^1^**		1.10 [1.09–1.12]	0.09	62	0.01 *
**MD at diagnosis ^1^**		0.77 [0.67–0.9]	0.24	71	5 × 10^−4^ *
**VFI at diagnosis ^1^**		0.93 [0.89–0.98]	0.25	70	3 × 10^−3^ *
***LOXL1*_ ** **rs2165241 ^1^**		6.2 [3.06–12.56]	0.43	76	4 × 10^−7^ *
***LOXL1*_ ** **rs1048661 ^1^**		2.6 [1.54–4.57]	0.19	65	3 × 10^−4^ *
**IOP * MD at diagnosis ^1^**		0.99 [0.98–1]	0.32	72	0.04 *
**IOP * VFI at diagnosis ^1^**		N.A.	N.A.	N.A.	0.24
***LOXL1*_ ** **rs2165241 * IOP at diagnosis ^1^**		1.05 [1.03–1.08]	0.45	77	3 × 10^−8^ *
***LOXL1*_ ** **rs1048661 * IOP at diagnosis ^1^**		1.03 [1.01–1.05]	0.24	65	5 × 10^−5^ *
***LOXL1*_ ** **rs2165241 * MD at diagnosis ^1^**		0.69 [0.59–0.81]	0.54	80	7 × 10^−6^ *
***LOXL1*_ ** **rs1048661 * MD at diagnosis ^1^**		0.82 [0.74–0.9]	0.36	74	1 × 10^−4^ *
***LOXL1*_ ** **rs2165241 * VFI at diagnosis ^1^**		1.02 [1.01–1.03]	0.34	78	5 × 10^−6^ *
***LOXL1*_ ** **rs1048661 * VFI at diagnosis ^1^**		1.01 [1–1.02]	0.26	72	4 × 10^−3^ *

GPA: Guided glaucoma progression. PA: Prediction accuracy (classification table). MD: Mean deviation. VFI: Visual field index. IOP: Intraocular pressure. SLT: Selective laser trabeculoplasty. ^1^ Adjusted for SLT treatment and the number of medications. ^(2)^ Adjusted R^2^: Nagelkerke R square. The * are the significants values.

**Table 3 biomedicines-12-01225-t003:** Summary of the results obtained from the analysis of the area under the curve (AUC).

	AUC [95% CI]	*p*	Sensitivity	Specificity
**IOP at diagnosis**	0.70 [0.59–0.82]	3 × 10^−4^	0.70	0.67
***LOXL1*_ ** **rs2165241**	0.80 [0.72–0.90]	1.6 × 10^−8^	0.92	0.58
**IOP at diagnosis * *LOXL1*_rs2165241**	0.85 [0.77–0.92]	2.3 × 10^−12^	0.80	0.73
***LOXL1*_ ** **rs1048661**	0.71 [0.60–0.81]	1.8 × 10^−4^	0.78	0.55
**IOP at diagnosis * *LOXL1*_rs1048661**	0.77 [0.67–0.86]	4 × 10^−8^	0.75	0.71
**MD at diagnosis**	0.8 [0.7–0.89]	1.5 × 10^−8^	0.78	0.76
**MD at diagnosis * *LOXL1*_rs2165241**	0.90 [0.84–0.96]	2.2 × 10^−14^	0.82	0.64
**MD at diagnosis * *LOXL1*_rs1048661**	0.82 [0.73–0.90]	5.4 × 10^−13^	0.78	0.76
**VFI at diagnosis**	0.79 [0.69–0.88]	8.5 × 10^−9^	0.75	0.70
**VFI at diagnosis * *LOXL1*_rs2165241**	0.91 [0.85–0.97]	3.2 × 10^−15^	0.82	0.65
**VFI at diagnosis * *LOXL1*_rs1048661**	0.81 [0.70–0.88]	3.7 × 10^−10^	0.78	0.64

IOP: Intraocular pressure. MD: Mean deviation. VFI: Visual field index.

## Data Availability

Dataset available on request from the author.

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
