# Peer review of "Adding Genetics to the Risk Factors Model Improved Accuracy for Detecting Visual Field Progression in Newly Diagnosed Exfoliation Glaucoma Patients"

_biomedicines, 2024, doi:10.3390/biomedicines12061225_

Round 1

Reviewer 1 Report

Comments and Suggestions for Authors

Excellent work, just one comment: Please describe which device you used to measure IOP

Author Response

Thanks for your comment. The required information was now added to the manuscript. 

Reviewer 2 Report

Comments and Suggestions for Authors

The authors asked the question, if inclusion of genetics (testing of 2 SNPs of the LOXL-1-gene) as an additional risk factor for progression would improve the accuracy of the models used in newly diagnosed exfoliation glaucoma (early and moderate) patients in a prospective cohort study. The increase of the AUC-values of the models was of around 10-20% when these 2 SNPs were included. This result will help clinicians to better manage patients with exfoliation glaucoma.

In summary interesting to read, of clinical value, perfect interpretation of statistics and results, Figures and Tables informative. Excellent paper.

[Just 1 comment: line 43: authors instead of author.]

Author Response

Thanks for your comments. A small change in "author" to "authors" has been made. 

Reviewer 3 Report

Comments and Suggestions for Authors

The authors aimed to show if including genetics as a risk factor for progression will improve the accuracy of the models used in newly diagnosed exfoliation glaucoma patients. The subject of this study is quite interesting. However, several points and questions need to be explained and answered.

Because the introduction section appeared to be too long, the paragraph starting ‘'Researchers often use regression analysis~’’ regarding the regression analysis could be removed.

In discussion, the authors needed to comment the cost effectiveness of the KASPar PCR SNP genotyping system, when we followed-up the exfoliation glaucoma patients. Especially, in the underdeveloped or developing country, such a system cannot be available. They had better commenting and comparing the several methods to detect these SNPs.

Comments on the Quality of English Language

The quality of English language was fine. 

Author Response

Dear reviewer, thanks for your comments. 

I deleted the paragraph in the introduction section as required. 

I included the information about the cost-effectiveness of this analysis in the discussion part under the limitations of the study.